# CRISPR-Cas9 in Cardiovascular Medicine: Unlocking New Potential for Treatment

**DOI:** 10.3390/cells14020131

**Published:** 2025-01-17

**Authors:** Klaudia Bonowicz, Dominika Jerka, Klaudia Piekarska, Janet Olagbaju, Laura Stapleton, Munirat Shobowale, Andrzej Bartosiński, Magdalena Łapot, Yidong Bai, Maciej Gagat

**Affiliations:** 1Department of Histology and Embryology and Vascular Biology Student Research Club, Collegium Medicum in Bydgoszcz, Nicolaus Copernicus University in Torun, 85-092 Bydgoszcz, Poland; klaudia.bonowicz@cm.umk.pl (K.B.); dominika.jerka@cm.umk.pl (D.J.); klaudia.mikolajczyk.piekarska@gmail.com (K.P.); olagbaju12344@gmail.com (J.O.); stapes.laura@gmail.com (L.S.); doyin5724@gmail.com (M.S.); 2Faculty of Medicine, Collegium Medicum, Mazovian Academy in Płock, 09-402 Płock, Poland; a.bartosinski@mazowiecka.edu.pl (A.B.); m.lapot@mazowiecka.edu.pl (M.Ł.); 3Department of Cell Systems and Anatomy, UT Health, Long School of Medicine, San Antonio, TX 78229, USA; baiy@uthscsa.edu

**Keywords:** cardiovascular diseases (CVDs), CRISPR-Cas9, gene therapy, mitochondrial genome editing

## Abstract

Cardiovascular diseases (CVDs) remain a significant global health challenge, with many current treatments addressing symptoms rather than the genetic roots of these conditions. The advent of CRISPR-Cas9 technology has revolutionized genome editing, offering a transformative approach to targeting disease-causing mutations directly. This article examines the potential of CRISPR-Cas9 in the treatment of various CVDs, including atherosclerosis, arrhythmias, cardiomyopathies, hypertension, and Duchenne muscular dystrophy (DMD). The technology’s ability to correct single-gene mutations with high precision and efficiency positions it as a groundbreaking tool in cardiovascular therapy. Recent developments have extended the capabilities of CRISPR-Cas9 to include mitochondrial genome editing, a critical advancement for addressing mitochondrial dysfunctions often linked to cardiovascular disorders. Despite its promise, significant challenges remain, including off-target effects, ethical concerns, and limitations in delivery methods, which hinder its translation into clinical practice. This article also explores the ethical and regulatory considerations surrounding gene editing technologies, emphasizing the implications of somatic versus germline modifications. Future research efforts should aim to enhance the accuracy of CRISPR-Cas9, improve delivery systems for targeted tissues, and ensure the safety and efficacy of treatments in the long term. Overcoming these obstacles could enable CRISPR-Cas9 to not only treat but also potentially cure genetically driven cardiovascular diseases, heralding a new era in precision medicine for cardiovascular health.

## 1. Introduction

CVD remains a leading cause of morbidity and mortality worldwide, affecting both developed and developing countries [1]. It is strongly associated with reduced life expectancy and significantly diminished quality of life. It encompasses a broad spectrum of disorders affecting the heart and blood vessels, including stroke, heart failure, hypertensive crises, and various other vascular and cardiac pathologies. Between 1990 and 2015, mortality due to CVD increased by 42.4%, underscoring its growing global burden [2]. CVD is closely correlated with key risk factors, including diabetes, obesity, physical inactivity, and an unhealthy diet [3]. In addition to its severe health consequences, CVD greatly diminishes the quality of life, often leading to disability, premature retirement, and a reduced ability to perform daily tasks, which can negatively affect workplace productivity. The burden of CVD extends beyond individual patients, exerting substantial pressure on healthcare systems. In 2003, the economic impact of CVD on 25 EU countries was estimated at EUR 169 billion, a figure that rose to EUR 210 billion by 2017 with the inclusion of 28 member states. This growing financial strain underscores the urgent need for effective prevention and management strategies [4].

There is a clear desire for innovative treatment in this field largely due to its increasing prevalence in almost all countries, contrary to the vigorous efforts at prevention. Modernizing CVD treatment through the innovative application of knowledge and technology remains an achievable and urgent goal [5]. Over time, significant progress has been made in shifting the focus from symptom-based approaches to prevention-centered strategies, particularly in addressing risk factors such as hypertension and hyperlipidemia. However, despite these advancements, CVD continues to be the leading cause of death worldwide, with the absolute number of fatalities rising due to population growth and aging. Encouragingly, age-standardized mortality rates from CVD have decreased in many high-income countries, largely due to advancements in early diagnosis, treatment, and risk factor management [6]. To combat the ever-rising CVD risk factors, including type two diabetes and obesity, it is necessary more than ever for innovative new technology to help shift a decline in CVD development [7].

CRISPR-Cas9 systems are currently recognized as the most efficient programmable nucleases for precisely inducing double-strand breaks (DSBs) at targeted genomic sites. Their high specificity and versatility have revolutionized genome editing, allowing for precise modifications in various organisms and cell types [8]. The system’s simplicity and efficacy in achieving desired outcomes make it a valuable tool in genetic research and therapy. An important consideration in CRISPR-Cas9 applications is the distinction between somatic and germline editing. Somatic editing targets non-reproductive cells, affecting only the treated individual, while germline editing involves modifications in reproductive cells or embryos, making changes heritable. Although germline editing holds the potential for preventing heritable diseases, it raises ethical concerns due to risks of unintended consequences and implications for future generations [9].

The CRISPR-Cas9 complex consists of the Cas9 protein, which acts as a nuclease capable of cleaving both DNA strands and a single guide RNA (sgRNA) that directs Cas9 to the targeted genomic location. In addition to its applications in nuclear DNA, recent advances have extended CRISPR-Cas9’s potential to mitochondrial DNA, offering new therapeutic possibilities, particularly in cardiovascular diseases where mitochondrial dysfunction, coupled with the chronification of inflammation and specific mitochondrial mutations, plays a significant role [10,11]. Moreover, epigenome editing—a unique application of this system—differs from conventional gene editing approaches by altering gene expression without directly modifying the DNA sequence. Instead, it affects the interaction between proteins and DNA sequences, thereby regulating gene activity [12]. This form of editing has significant implications for the treatment of CVD, as the genetic basis of many cardiovascular conditions is well established [13]. The ability to identify and target specific genes related to CVD offers a promising avenue for integrating CRISPR-Cas9 technologies into therapeutic strategies aimed at addressing cardiovascular diseases [14].

## 2. Structure and Function of CRISPR-Cas9: Key Components, Mechanisms, and Optimization Strategies

CRISPR-Cas9 technology has made a profound impact, earning widespread recognition for revolutionizing molecular biology and driving advancements in genetic research [15,16]. The gRNA directs Cas9 to a specific DNA sequence, enabling precise genome editing. Typically designed as a single-guide RNA (sgRNA) in experimental applications, the gRNA consists of a scaffold region that binds to Cas9 and a customizable guide sequence complementary to the target DNA. The evolutionary origin of CRISPR-Cas9 as part of the bacterial immune system, developed to degrade foreign genetic material during viral attacks, highlights its inherent specificity and versatility for genetic engineering [17].

The Cas9 protein functions as a “molecular scissor”, creating double-strand breaks (DSBs) at specific target sites in the DNA. Structurally, Cas9 contains two nuclease domains—RuvC and HNH—responsible for cleaving the non-complementary and complementary DNA strands, respectively. Additionally, a recognition domain binds the gRNA, ensuring precise targeting of the desired DNA sequence. When the gRNA and Cas9 interact, conformational changes occur within the protein, enhancing its capacity to bind and cleave DNA accurately. These structural shifts act as checkpoints to minimize off-target activity, which is critical for achieving high specificity [18,19]. The requirement for a protospacer adjacent motif (PAM) near the target site ensures that Cas9 does not indiscriminately bind DNA, serving as an additional layer of specificity critical for distinguishing between target and non-target sequences. To prevent self-targeting in bacterial cells, the system relies on the absence of PAM sequences in the host genome, a safeguard that directs Cas9 activity exclusively to foreign DNA [17,20].

The CRISPR-Cas9 genome editing process involves the precise delivery of its key components—Cas9 protein and guide RNA (gRNA)—into target cells. This delivery can be achieved through viral vectors, such as adenoviruses, which provide efficient cellular uptake and sustained expression, or via direct injection methods, including electroporation or microinjection, which introduce the editing machinery directly into the cytoplasm or nucleus to facilitate genome modification [21,22]. The commonly used delivery systems relevant to this study are summarized in Table 1. The synergy between the gRNA and the Cas9 protein underpins the success of this process. The gRNA, designed to include a sequence complementary to the target DNA, mimics the natural bacterial CRISPR-Cas9 system that directs nucleases to foreign viral DNA. By forming a complex with Cas9, the gRNA guides the protein to the specific genomic sequence, where Cas9 cleaves both DNA strands at the target site, initiating the editing process. This molecular interaction highlights the precision and versatility of the CRISPR-Cas9 system in genome manipulation [23,24].

### Exploring the Role of Guide RNA and Cas9 Protein in Genome Editing

The CRISPR-Cas9 system achieves precise genome editing through the coordinated function of the gRNA and the Cas9 protein. The gRNA’s 5′ CRISPR-Cas9 RNA (crRNA) region plays a pivotal role by enabling complementary base pairing with the target DNA sequence, ensuring accurate guidance for Cas9. The gRNA, as part of its natural form in prokaryotic systems, consists of two separate RNA components: the crRNA and the trans-activating CRISPR-Cas9 RNA (tracrRNA). The crRNA is derived from virus-derived DNA fragments, which enable sequence-specific targeting, while the tracrRNA interacts with Cas9 to form a functional complex. For genome editing purposes, these two RNA components are typically engineered into a sgRNA, simplifying the system and enhancing its utility. The sgRNA combines the sequence-specificity of the crRNA and the structural scaffolding role of the tracrRNA, ensuring compatibility with Cas9 and maintaining high levels of editing efficiency [31,32]. This interaction is contingent on the presence of a PAM downstream of the target site. The PAM sequence, typically 5′-NGG-3′ (where “N” represents any nucleotide), serves as a key molecular signal for Cas9 to locate and bind the correct DNA region. Upon recognizing the PAM, Cas9 induces local DNA melting, forming a stable RNA-DNA hybrid that activates its nuclease function. The protein’s HNH domain cleaves the complementary strand of DNA, while the RuvC domain targets the non-complementary strand, resulting in a DSB with blunt ends. This mechanism underpins the precision of CRISPR-Cas9 as a genome editing tool. Although Cas9 remains the most widely used nuclease, alternative enzymes, such as Cpf1 (Cas12a), have emerged, offering unique features like distinct PAM requirements and staggered DNA cuts, which expand the range of targetable genomic regions [33].

Following the creation of a DSB, the cell’s DNA repair machinery comes into play, utilizing either non-homologous end joining (NHEJ) or homology-directed repair (HDR). NHEJ is often error-prone, introducing small insertions or deletions that can disrupt gene function, making it suitable for knockout applications. In contrast, HDR leverages a donor DNA template to enable precise gene correction or insertion, facilitating targeted genetic modifications (Figure 1). These repair pathways are essential for realizing the full potential of CRISPR-Cas9 in both research and therapeutic contexts. A significant challenge in CRISPR-based editing is the minimization of off-target effects, which can compromise editing accuracy and introduce unintended mutations. Strategies to enhance gRNA design and optimize experimental protocols are crucial for improving specificity and reducing off-target activity. Moreover, while the mechanism of Cas9-induced DNA melting remains incompletely understood, it represents a promising area for refining editing precision. Advances in sgRNA engineering and the development of alternative nucleases like Cpf1 continue to broaden the versatility and efficiency of the CRISPR-Cas9 toolkit, offering new opportunities for genome manipulation in diverse applications [34].

## 3. CRISPR-Cas9 Applications in Cardiovascular Disease Treatment

### 3.1. Overview of CRISPR-Cas9 in CVD

CRISPR-Cas9 has opened new avenues for translational medicine, allowing discoveries in basic research to be applied in clinical settings. While advancements in understanding gene functions have revolutionized therapeutic strategies, challenges such as off-target effects and limitations in editing specific sites have posed obstacles to gene editing technologies. Fortunately, the development of CRISPR-Cas9 offers new promise. This versatile system enables precise gene editing functions, making it well-suited for applications in translational medicine, particularly for genomic modifications required in clinical interventions [35]. 

Genetic factors contribute significantly to many cardiovascular conditions, making them prime candidates for CRISPR-Cas9-based interventions. Advanced atherosclerosis is the underlying cause of coronary artery disease (CAD), with heritability estimates ranging from 40% to 70%, suggesting strong genetic contributions to disease pathology [36]. Genetic variations, such as mutations and common polymorphisms, play a role in modulating risk factors like plasma lipoprotein levels, inflammation, and vascular calcification, consistent with findings from twin studies suggesting that the heritability of coronary atherosclerosis, based on fatal cardiac events, ranges from 38% to 57% [37]. Familial hypercholesterolemia (FH), the most common monogenic disorder leading to premature CVD, affects approximately 1 in 200 individuals and is primarily caused by mutations in the LDLR gene. Among over 2900 identified LDLR variants, around 1000 are pathogenic. Similarly, hypertrophic cardiomyopathy (HCM) affects 1 in 500 individuals and is linked to mutations in over 11 sarcomeric genes, with more than 1400 variants identified [38]. In turn, most arrhythmia syndromes are inherited in an autosomal dominant manner, with first-degree relatives having a 50% chance of inheriting the condition [39]. Genetic factors also play a significant role in hypertension, a key risk factor for cardiovascular diseases. The AGT gene, which encodes angiotensinogen, is among the most extensively studied contributors to blood pressure regulation. Variants in this gene are linked to elevated angiotensinogen levels and a higher risk of hypertension. With heritability estimates for blood pressure ranging from 30% to 50%, genetic interventions hold substantial potential for addressing this condition [40]. Importantly, approximately 40% of individuals with mitochondrial diseases experience cardiovascular complications, including cardiomyopathy and arrhythmias. Among these, cardiomyopathy is particularly prevalent, affecting 20–25% of adults with mitochondrial disorders. Additionally, arrhythmias and conduction defects, such as atrial fibrillation and AV block, are observed in 26% of cases, underscoring the significant role of mitochondrial genetic defects in the development of cardiovascular diseases [41]. 

Genomic editing offers the potential to directly correct single-gene mutations responsible for certain CVDs, presenting a promising avenue for treatment and even potential eradication of specific forms of the disease [42]. Additionally, novel approaches that utilize CRISPR-Cas9 for transient protein modulation rather than permanent genome editing further expand its therapeutic potential. The feasibility of applying CRISPR-Cas9 technology in human systems has been demonstrated through successful zygote injections of beneficial alleles into early human embryos [43]. This section delves into key applications of CRISPR-Cas9, focusing on repairing genetic mutations, modulating protein expression, and addressing mitochondrial dysfunction. 

### 3.2. Repairing Genetic Mutations

A key application of CRISPR-Cas9 in CVD treatment is correcting disease-causing genetic mutations [44]. Some forms of CVD follow a straightforward inheritance pattern, indicating a single causative gene with a significant impact on the phenotype. Familial hypercholesterolemia (FH), caused by mutations in the low-density lipoprotein receptor (LDLR) gene, exemplifies this, making LDLR a prime target for genome editing [45]. The pivotal role of LDL in atherosclerosis development highlights the therapeutic potential of reducing LDL levels to lower CVD risk. Strategies aimed at lowering LDL and triglycerides while increasing HDL levels remain central in addressing atherosclerosis, a chronic inflammatory disease affecting more than 25% of adults globally and a leading cause of death [46,47]. Characterized by fibrofatty plaques in arterial walls, atherosclerosis underpins severe conditions like ischemic stroke, myocardial infarction, heart failure, and peripheral artery disease [48,49,50]. One relevant mutation associated with FH is LdlrE208X, corresponding to the human E207X. This nonsense mutation introduces a premature stop codon, resulting in non-functional LDL receptors, impaired LDL clearance, and accelerated atherosclerosis. Zhao et al. demonstrated that delivering CRISPR-Cas9 via AAV8 to a mouse model with this mutation partially restored LDL receptor expression and improved atherosclerotic phenotypes. Treated mice showed reductions in total cholesterol, LDL cholesterol, and triglyceride levels, along with decreased macrophage infiltration and smaller plaques. Notably, no significant off-target effects were detected, underscoring the safety of this approach [51].

CRISPR-Cas9 genome editing has shown promise in the treatment of familial cardiomyopathies, including dilated cardiomyopathy (DCM) and hypertrophic cardiomyopathy (HCM). DCM, characterized by an enlarged and weakened heart muscle, is one of the leading causes of heart failure, with an estimated prevalence of 1 in 250 individuals. This condition often leads to impaired cardiac function and increased mortality [52]. A significant advancement in treating DCM involves the use of the ABEmax-VRQR-SpCas9 system, an optimized base-editing variant of CRISPR-Cas9. This technology was applied to correct a mutation in the RBM20 gene, specifically the RBM20R634Q and RBM20636s mutations, which are strongly associated with DCM. The correction of these mutations in induced pluripotent stem cells (iPSCs) and subsequent application in murine cardiac tissues resulted in remarkable outcomes. Treated subjects exhibited a reversal of cardiac dilation, restoration of normal cardiac function, and significant improvements in life span. In stark contrast, untreated mice displayed severe ventricular and atrial enlargement, hallmark features of DCM [53].

CRISPR-Cas9 has also been employed to address mutations in the TTN gene, which encodes titin, a key protein involved in the structure and function of sarcomeres—the fundamental contractile units of cardiac muscle. Truncating mutations in the A-band region of the TTN gene (A-band-TTNtv) disrupt titin production, leading to reduced cardiac contractility. Using CRISPR-Cas9 in human-induced pluripotent stem cell-derived cardiomyocytes (hiPSC-CMs), researchers successfully restored titin production, significantly improving contractile function. These findings highlight the potential of genome editing to address structural and functional deficits in cardiomyopathies [54]. Beyond its therapeutic applications, CRISPR-Cas9 has been leveraged as a screening tool to identify critical host factors in the pathogenesis of viral myocarditis—a condition that can precipitate DCM. For instance, researchers identified ADAM9, a metalloproteinase involved in extracellular matrix remodeling, as a key factor in the early stages of encephalomyocarditis virus (EMCV) infection in both human and murine models. This discovery sheds light on the molecular mechanisms of viral invasion and provides a foundation for developing novel strategies to prevent viral myocarditis, a significant contributor to DCM [55].

CRISPR-Cas9 genome editing has revolutionized the treatment landscape for familial cardiomyopathies and other inherited cardiac conditions. One notable success is in addressing hypertrophic cardiomyopathy (HCM), a condition caused by pathogenic variants such as the R403Q mutation in the myosin heavy chain gene (MYH6). HCM is characterized by thickening of the heart muscle, which can impede normal cardiac function. Using adeno-associated virus serotype 9 (AAV9) to deliver guide RNA (sgRNA) specifically targeting the MYH6 locus in cardiomyocytes, researchers achieved remarkable results [56]. A single AAV9 delivery rendered the pathogenic allele inactive in more than 70% of ventricular myocytes. This intervention effectively prevented the structural and functional hallmarks of HCM in treated mice, preserving normal heart morphology and function [57]. Expanding on this, Noonan syndrome—a genetic condition associated with HCM caused by mutations in the LZTR1 gene—has also been targeted using CRISPR-Cas9 technology. In this case, CRISPR-mediated editing of patient-derived iPSCs corrected the LZTR1 mutations, restoring normal cellular function in cardiomyocytes and reversing hypertrophic characteristics. These results further underscore the transformative potential of CRISPR-CAS9 in treating patient-specific genetic defects [58]. CRISPR-Cas9 technology has also been applied in embryonic contexts to correct pathogenic mutations associated with inherited cardiomyopathies, emphasizing the potential of early intervention. One notable example involves mutations in the MYBPC3 gene, a common cause of hypertrophic cardiomyopathy. Using CRISPR-Cas9, researchers achieved precise correction of a heterozygous 4 bp deletion in exon 16 of MYBPC3 with a success rate exceeding 70%. By delivering sgRNA, Cas9 protein, and a single-stranded oligonucleotide DNA (ssODN) into fertilized oocytes, the deletion was corrected via HDR, with no detectable mosaicism in the resulting embryos. This early intervention demonstrates the promise of correcting inherited mutations at the zygotic stage, potentially preventing the onset of disease before birth [29].

Similarly, CRISPR-Cas9 has demonstrated its potential in addressing PRKAG2 cardiac syndrome, an autosomal dominant disorder characterized by ventricular tachycardia and progressive heart failure [59]. By employing CRISPR-Cas9 to knock out the mutant PRKAG2 allele, researchers successfully mitigated the effects of this syndrome in mouse models [60]. Additionally, targeting the R176Q allele in the ryanodine receptor type 2 (RYR2) gene using AAV9-CRISPR-Cas9 prevented the onset of catecholaminergic polymorphic ventricular tachycardia (CPVT), a severe arrhythmia associated with this mutation [61]. 

In another groundbreaking application, researchers tackled mutations in the phospholamban (PLN) gene, specifically the R14del mutation, which disrupts calcium regulation in heart muscle cells and is linked to cardiomyopathy. Using AAV9-CRISPR-Cas9 to disrupt the mutant allele, researchers restored cardiac function and reduced the risk of sustained ventricular tachycardia in young adult mice carrying the mutation. These findings highlight the potential of genome editing to correct inherited cardiac diseases by targeting specific pathogenic alleles [62]. 

CRISPR-Cas9 has also shown promise in treating Duchenne muscular dystrophy (DMD), a severe genetic disorder caused by mutations in the dystrophin (DMD) gene on the X chromosome [63]. DMD is characterized by progressive muscle weakness, including cardiomyopathy, which is a leading cause of mortality [64]. Using a combination of AAV9-sgRNA, activation-induced cytidine deaminase (AID), and a modified CRISPR-Cas9 system, researchers induced exon skipping in the DMD gene, restoring up to 90% of dystrophin protein expression in the hearts of treated mice. This intervention halted early ventricular remodeling and significantly improved both cardiac and skeletal muscle function [65]. In cases involving exon 44 deletions in the DMD gene, CRISPR-Cas9-mediated editing components successfully restored dystrophin expression to 90% within just four weeks. This demonstrates the rapid and profound impact of genome editing on genetic disorders like DMD, providing hope for future clinical applications [66].

### 3.3. Reducing Protein Expression

Beyond genetic mutation correction, CRISPR-Cas9 can modulate protein expression, offering novel therapeutic pathways [67]. Recent research highlights its potential in addressing hypertension and atherosclerosis by targeting genes involved in protein overexpression. Hypertension, one of the most common chronic conditions globally, is closely linked to overexpression of the angiotensinogen (AGT) gene. AGT is a precursor protein in the renin-angiotensin system, which regulates blood pressure and fluid balance. Overactivity of this pathway contributes to sustained hypertension [68,69]. Targeting AGT using AAV8-Cas9-AGT gRNA (a viral vector delivering gene editing components) to disrupt exon 2 in liver hepatocytes resulted in significant and sustained reductions in systolic and diastolic blood pressure in spontaneously hypertensive adult rats. Importantly, partial ablation (40%) of AGT expression was sufficient to prevent hypertension without affecting the cardiovascular stress response, demonstrating the potential for precise and safe gene editing interventions [70]. 

Another approach involved targeting the G6PD (glucose-6-phosphate dehydrogenase) gene. Loss-of-function mutations in G6PD, introduced via crRNA/tracrRNA/Cas9 with a single-stranded oligodeoxynucleotide (ssODN), conferred protection against hypertension. Mutant rats (G6PDS188F) showed reduced arterial stiffness, improved arterial elasticity, and lower blood pressure metrics even under a high-fat diet [71]. However, the role of G6PD deficiency in cardiovascular health remains complex. While it may reduce atherogenesis, it is also associated with increased oxidative damage and worsened outcomes in some contexts, particularly in hematological diseases [72,73]. Additionally, CRISPR-Cas9 has been used to target the GPER1 (G protein-coupled estrogen receptor 1) gene, which plays a role in regulating vascular tone and blood pressure. Targeted deletion of a single exon in the GPER1 gene using multiplexed guide RNAs in salt-sensitive rats led to significant reductions in systolic and diastolic blood pressure, mean arterial pressure, and pulse pressure, underscoring the therapeutic potential of CRISPR-Cas9 for modulating vascular tone in hypertension [74].

In atherosclerosis, overexpression of specific proteins involved in lipid metabolism contributes to disease progression. One such protein, APOC3 (apolipoprotein C-III), inhibits lipoprotein lipase, preventing the clearance of triglyceride-rich lipoproteins [75]. Elevated APOC3 levels are strongly associated with hypertriglyceridemia and increased atherosclerotic risk [76]. Using a single guide RNA to disrupt exon 2 of APOC3 in rabbits resulted in lower plasma triglycerides, higher levels of high-density lipoprotein cholesterol (HDL-C), and reduced aortic plaque formation [77]. Similar results were observed in hamsters, where APOC3 deletion significantly decreased circulating triglyceride and cholesterol levels while markedly increasing HDL-C [78]. 

Another promising target for reducing LDL cholesterol levels is PCSK9 (proprotein convertase subtilisin/kexin type 9), a liver-secreted protein that regulates the degradation of LDLR. Elevated PCSK9 levels impair LDL clearance by decreasing the availability of LDL receptors on hepatocytes, contributing to hypercholesterolemia and cardiovascular disease risk. Recent preclinical studies have shown that CRISPR-Cas9 genome editing can effectively disrupt the Pcsk9 gene in mouse models. Ding et al. used adenovirus to deliver Cas9 and a guide RNA targeting exon 1 of the Pcsk9 gene directly into mouse liver cells, achieving a mutagenesis rate of over 50% within four days. This intervention resulted in a 35–40% reduction in plasma cholesterol levels, increased hepatic LDL receptor expression, and lower plasma PCSK9 levels without significant off-target effects. These findings demonstrate that PCSK9 disruption via CRISPR-Cas9 is a promising approach to improving cholesterol metabolism and reducing the risk of cardiovascular diseases [25]. 

Innovative delivery systems such as PuPGEA, a non-viral nanosystem based on pullulan, have further improved the efficiency and specificity of CRISPR-Cas9 editing. PuPGEA facilitates the delivery of the pCas9-Pcsk9 plasmid to liver cells, minimizing off-target effects in non-target organs [79]. In cynomolgus monkeys, a single administration of a base editor protein combining SpCas9 with an engineered TadA (tRNA adenosine deaminase) reduced PCSK9 protein levels by 83% and LDL-C by 69%, with lasting effects observed for up to 476 days. Notably, no evidence of germline editing was detected, underscoring the safety of this approach [80].

### 3.4. Targeting Mitochondrial Dysfunction

Mitochondrial dysfunction is a well-recognized contributor to cardiovascular diseases (CVD), making mitochondrial DNA (mtDNA) an appealing target for therapeutic intervention. Unlike nuclear DNA, mtDNA is maternally inherited and encodes essential proteins involved in oxidative phosphorylation, the primary process of cellular energy production. Mutations in mtDNA can impair energy metabolism, contributing to the onset of conditions such as hypertrophic cardiomyopathy, heart failure, and ischemic heart disease. While advancements in genome editing technologies have unlocked potential solutions for targeting nuclear DNA mutations, addressing mtDNA mutations remains in its infancy, with limited studies available to date. This nascent field offers an exciting yet challenging frontier for therapeutic innovation [81].

Recent studies have explored the connection between mitochondrial mutations and chronic inflammation in the context of CVD. For example, a 2024 study by Sukhorukov et al. examined the m.15059G > A mutation in the MT-CYB gene using cytoplasmic hybrid (cybrid) cell lines. This mutation disrupts mitophagy, the process responsible for selectively degrading damaged mitochondria, which is crucial for maintaining cellular health [11]. The study revealed that cybrid cell lines with defective mitophagy exhibited elevated secretion of pro-inflammatory cytokines such as CCL2, IL8, and TNF, contributing to a chronic inflammatory state. These cells also demonstrated exaggerated inflammatory responses, with increased levels of IL6, IL1β, and IL8, when repeatedly stimulated with bacterial lipopolysaccharide (LPS). Such findings underscore the link between mitochondrial dysfunction, chronic inflammation, and the progression of atherosclerosis [82]. By employing CRISPR-Cas9-based mitochondrial genome editing, researchers successfully eliminated mtDNA copies carrying the m.15059G > A mutation in these cybrids, restoring mitophagy and reducing pro-inflammatory responses. These results align with previous studies indicating that macrophage dysfunction due to impaired mitophagy is a key driver of chronic inflammation in atherosclerosis. The inability of these macrophages to regulate inflammation after repeated stimulation highlights the importance of mitochondrial function in maintaining immune tolerance. Given the central role of macrophages and inflammation in atherosclerosis, detailed focus on this topic is warranted, as it provides critical insights into the potential impact of mitochondrial genome editing on atherosclerosis and related CVDs [11].

Despite these promising advancements, the delivery of guide RNA (gRNA) to mitochondria remains a significant challenge. Unlike nuclear genome editing, mitochondrial genome editing requires gRNA to cross the double mitochondrial membrane, a formidable barrier. Current research efforts have focused on modifying gRNA with mitochondrial localization signals (MLSs) or incorporating stem-loop motifs to enhance mitochondrial import. While these modifications have shown some success, their practical application remains inconsistent. Developing robust delivery mechanisms will be essential for the wider application of CRISPR-Cas9 in treating mitochondrial disorders and related cardiovascular diseases [30]. 

The various applications of CRISPR-Cas9 technology in cardiovascular diseases are summarized and presented in Table 2.

## 4. Advantages and Challenges of CRISPR-Cas9 in Clinical Applications

CRISPR-Cas9 technology has become one of the most widely utilized gene editing tools in molecular biology labs worldwide due to its ease of design, low cost, rapid execution, and superior accuracy and efficiency compared to earlier methods [83]. Its ability to create genetically modified animal and cell models, such as gene knockout models, has proven invaluable for advancing modern medicine and paving the way for future treatments of currently incurable diseases [65,84]. It has also demonstrated immense promise for clinical applications, with its unparalleled precision, efficiency, and cost-effectiveness have facilitated the correction of pathogenic mutations, such as LDLR, which are critical for managing conditions like hypercholesterolemia and atherosclerosis [85]. Furthermore, the ability of CRISPR-Cas9 to be integrated into delivery platforms like AAVs enhances its potential for targeted therapy, offering a transformative approach to treating inherited and acquired cardiovascular conditions [86]. However, the development of genome editing technologies is not without its challenges.

A major concern for implementing CRISPR-Cas9 for gene therapy is the relatively high frequency of off-target effects (OTEs), which have been observed at a frequency of ≥50% [23]. Several CRISPR-Cas9 variants have been developed by researchers to minimize off-target effects while maintaining efficacy, for example, SpCas9-HF1, a high-fidelity variant. The occurrence of off-target events has been suggested to arise due to the SpCas9-sgRNA complex possessing an excess of energy beyond the threshold necessary for its recognition of the designated DNA site [87]. Through disruption of contacts between the direct hydrogen bonds of 4 residues and the phosphate backbone of the DNA in SpCas9-HF1, a complete lack of off-target events was seen in comparison to wild-type SpCas9 [88].

The precise delivery of CRISPR-Cas9 components to target tissues presents another significant limitation. Strategies under development include lipid nanoparticles, viral vectors, and novel non-viral delivery systems tailored for specific tissue types, but all face challenges in delivering to muscle and the cardiovascular system [26]. Lipid nanoparticles have limited delivery efficiency and tend to accumulate in the liver and spleen rather than targeting heart or muscle cells, while cardiomyocytes lack specific markers for effective uptake. AAV-based methods, although promising for cardiac gene delivery, are hindered by pre-existing immunity, immune responses, non-specific tissue transduction, high production costs, and limited packaging capacity. Dual AAV systems require co-infection of the same cell, reducing efficiency, and high doses raise safety concerns. Across all methods, off-target effects, long-term expression risks, and endothelial resistance to transduction present additional hurdles, emphasizing the need for precise, tissue-specific, and safe delivery strategies [26]. The need to minimize toxicity adds another layer of complexity. Toxicity may arise from the delivery vehicle itself, immune responses, or unintended off-target effects, particularly when high doses are administered. To address these challenges, researchers are exploring biocompatible materials, such as pullulan-modified polymers, which can reduce systemic toxicity while enhancing targeting specificity. Additionally, tissue-specific delivery strategies and optimized dosage protocols are crucial to avoid unintended damage to non-target tissues, particularly in sensitive cardiovascular structures such as endothelial cells and cardiomyocytes [79].

Another critical consideration is the genomic stability of edited cells. While CRISPR-Cas9 is highly efficient in inducing targeted edits, studies have shown that it can also lead to unintended genomic changes, such as large deletions, complex rearrangements, and off-target effects. These alterations may have pathogenic consequences, particularly in mitotically active cells, where the risk of carcinogenic lesions or disruptions in regulatory elements is heightened. Furthermore, evidence suggests that loss of heterozygosity and extensive DNA damage can elicit long-range transcriptional consequences, potentially activating oncogenic pathways. Comprehensive assessments, including long-read sequencing and functional analyses, are essential to evaluate and mitigate these risks. To address these challenges, ensuring genomic stability through careful design of guide RNAs, optimized delivery methods, and stringent validation protocols is imperative [89,90]. Additionally, while chemical modifications improve sgRNA stability and reduce toxicity, further research is essential to understand their long-term impact on cellular viability and functionality, particularly in sensitive primary cells [91].

Immune responses to Cas9 proteins pose a significant challenge for CRISPR-based therapies, as many individuals harbor preexisting adaptive immunity to Staphylococcus aureus and Streptococcus pyogenes Cas9 orthologs, with high levels of anti-Cas9 antibodies and T cells. These immune responses can impair gene editing efficiency and increase the risk of immune-mediated side effects. To address these issues, strategies such as employing immune-orthogonal Cas9 variants, engineering proteins with reduced immunogenicity, and using transient immunosuppressive regimens have been explored [92]. Research demonstrates that leveraging the evolutionary diversity of Cas9 proteins enables the sequential use of immune-orthogonal orthologs, which can circumvent adaptive immune responses and allow repeated dosing without compromising efficacy. Experimental validation in murine models has confirmed that these orthologs avoid eliciting cross-reactive memory B- and T-cell responses, preserving gene editing efficiency [93]. These advancements emphasize the importance of refining protein engineering and delivery methods to overcome immune barriers and enhance the therapeutic potential of CRISPR-based technologies.

The use of CRISPR-Cas9 technology raises significant ethical concerns about safety, efficacy, and the consequences of genetic modification. While somatic editing has been permitted, embryonic editing has been met with heavy skepticism due to the unpredictable and permanent side effects that may be passed down through generations, calling into question the autonomy and consent of the later generations [94]. CRISPR-Cas9 applications, as well as associated advantages and challenges, are summarized in Figure 2.

## 5. Emerging Perspectives and Future Research Directions

CRISPR-Cas9 technology in cardiovascular disease research holds immense promise, particularly in addressing the genetic factors underlying cardiac disorders. While state-of-the-art applications focus on preclinical and early-phase studies, emerging perspectives delve into the untapped potential of this technology. These future directions emphasize refining techniques and tackling challenges that currently limit therapeutic applications. Unlike existing examples, these emerging avenues remain largely theoretical or in exploratory stages, guided by advancements in related fields. 

One area of future exploration involves the integration of CRISPR-Cas9 with next-generation sequencing to personalize interventions for inherited cardiovascular diseases. The ability to precisely edit pathological mutations within a patient’s genome introduces the possibility of curative therapies extending beyond conventional approaches. For example, correcting single-nucleotide mutations associated with hypertrophic cardiomyopathy could revolutionize treatment paradigms, provided challenges in repair mechanisms, such as balancing error-prone non-homologous end joining (NHEJ) and homology-directed repair (HDR), are resolved. Current studies have demonstrated feasibility in embryonic cells, but translating this to clinical-scale therapies necessitates overcoming issues of mosaicism, off-target effects, and ethical concerns [95].

Emerging applications also target CAD, shifting focus from symptom management to addressing root causes. CRISPR-Cas9 could inhibit PCSK9 to lower LDL levels, offering an alternative for patients unresponsive to statins. While animal models have validated this approach, significant challenges remain in adapting delivery systems for humans, particularly in ensuring specificity and avoiding immune responses. Future research could explore novel delivery platforms, such as lipid nanoparticles tailored for cardiovascular tissue, to enhance safety and efficacy [96,97]. 

Inherited arrhythmias, such as long QT syndrome, represent another promising application of CRISPR-Cas9. Current in vitro studies suggest the potential for gene correction in cardiomyocytes derived from induced pluripotent stem cells. However, translating this into in vivo applications will require robust solutions for efficient delivery and sustained gene expression. Additionally, combining genome editing with pharmacogenomics may enhance therapeutic precision, allowing for tailored interventions based on individual genetic profiles [98,99].

While these examples highlight promising avenues, emerging perspectives also include broader goals such as minimizing off-target effects, improving delivery mechanisms, and addressing ethical considerations surrounding germline editing. Tackling these challenges will be critical to transitioning CRISPR-Cas9 from experimental settings to mainstream clinical practice, ultimately enabling transformative therapies for cardiovascular diseases.

## 6. Summary

CRISPR-Cas9 technology offers an innovative and promising approach to treating CVDs by precisely targeting the genetic mutations underlying these conditions while also being employed to modulate the expression of genes that contribute to disease progression. Furthermore, CRISPR-Cas9 can be utilized to create disease models for understanding the pathophysiology of CVDs and validate engineer cell-based therapies, such as modifying iPSC-derived cardiomyocytes for personalized regenerative medicine. This gene editing system has demonstrated significant potential in addressing a wide range of cardiovascular issues, including atherosclerosis, arrhythmias, cardiomyopathies, hypertension, and DMD. Additionally, advancements in mitochondrial genome editing open new therapeutic possibilities, especially in diseases where mitochondrial dysfunction plays a key role. Despite these advances, CRISPR-Cas9 faces critical challenges, including off-target effects, ethical concerns, and the need for more refined methods of delivery. Addressing these limitations is crucial for translating laboratory success into clinical practice. Furthermore, as the technology moves closer to human application, long-term safety and efficacy must be rigorously evaluated. Future research should focus not only on improving the precision and reducing risks but also on developing robust delivery systems, particularly for mitochondrial genome editing. Ethical and regulatory frameworks will also need to evolve to address the broader implications of genome editing in both somatic and germline cells. In summary, CRISPR-Cas9 represents a groundbreaking tool for treating cardiovascular diseases, offering the potential to revolutionize cardiovascular therapy and significantly improve patient outcomes and quality of life. With continued advancements, this technology could fundamentally reshape how genetic diseases are treated.

## Figures and Tables

**Figure 1 cells-14-00131-f001:**
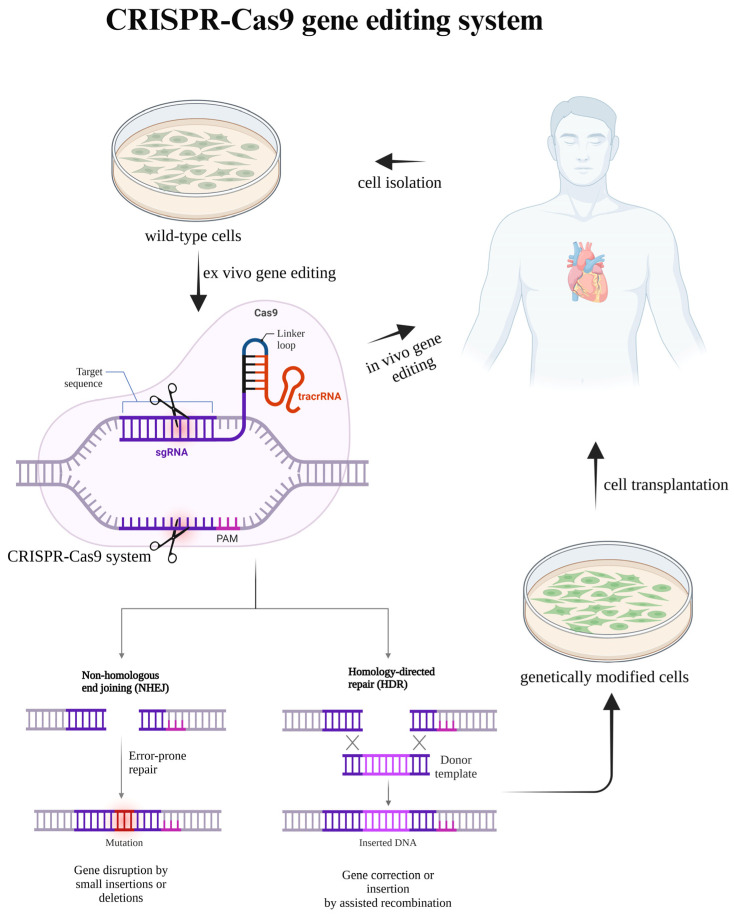
The mechanism of the CRISPR-Cas system. The CRISPR-Cas9 genome editing mechanism relies on DNA repair pathways, such as non-homologous end joining (NHEJ) and homology-directed repair (HDR), to restore genetic integrity after Cas9-induced double-strand breaks (DSBs). NHEJ often results in small insertions or deletions, while HDR allows for precise modifications using a repair template. Therapeutically, CRISPR-Cas9 can be applied through both ex vivo and in vivo approaches. In the ex vivo approach, patient-derived cardiovascular cells are genetically corrected outside the body and subsequently transplanted back into the same patient. Alternatively, in the in vivo approach, CRISPR-Cas9 components are directly delivered to target tissues to induce genetic corrections within the organism. The CRISPR-Cas9 system consists of Cas9 (purple), which introduces a double-strand break, sgRNA (dark purple) that guides Cas9 to the target DNA, and tracrRNA (red) that stabilizes the complex. The PAM sequence (pink) is essential for Cas9 recognition. Created in 2024 using BioRender.

**Figure 2 cells-14-00131-f002:**
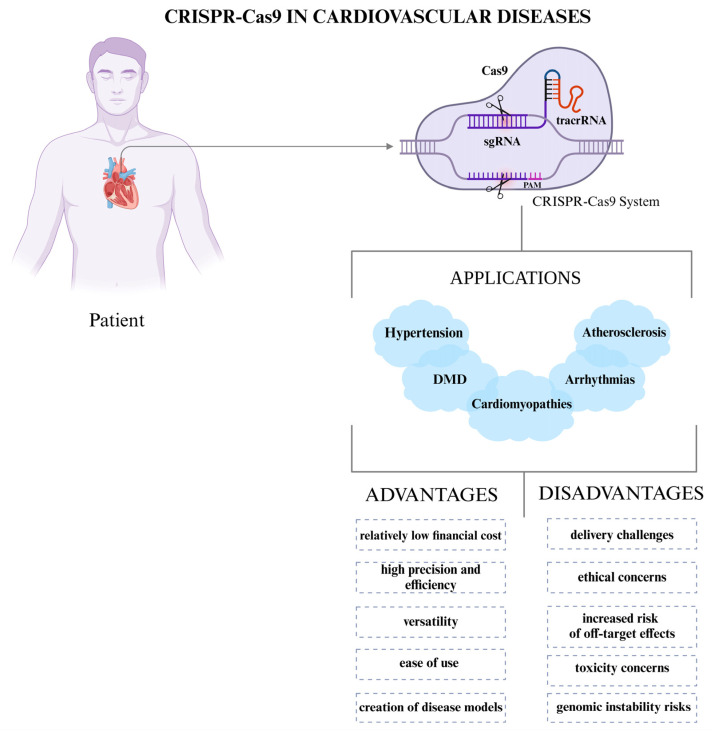
The use of CRISPR-Cas9 in cardiovascular diseases.CRISPR-Cas9 enables targeted gene editing through a system where sgRNA and tracrRNA guide the Cas9 nuclease to specific DNA sequences, creating double-strand breaks at the PAM site. It holds potential for treating conditions like hypertension, atherosclerosis, cardiomyopathies, arrhythmias, and Duchenne muscular dystrophy (DMD). Its advantages include relatively low cost, high precision, efficiency, versatility, ease of use, and the ability to generate disease models. However, its application faces challenges such as delivery difficulties, ethical concerns, off-target effects, toxicity risks, and potential genomic instability. The CRISPR-Cas9 system consists of Cas9 (purple), which introduces a double-strand break, sgRNA (dark purple) that guides Cas9 to the target DNA, and tracrRNA (red) that stabilizes the complex. The PAM sequence (pink) is essential for Cas9 recognition. This was created in 2024 using BioRender.

**Table 1 cells-14-00131-t001:** CRISPR-Cas9 delivery systems in cardiovascular research.

CRISPR-Cas9 Delivery Technology	Description	References
Adeno-Associated Viral Vectors (AAV)	AAVs are popular vectors for delivering CRISPR-Cas9 components, particularly to the heart. They enable specific DNA delivery but have limitations, such as low packaging capacity and the risk of immune responses. For example, they are used to edit PCSK9 genes to lower LDL cholesterol levels.	[25]
Dual AAV Systems	These systems split larger CRISPR-Cas9 components into two vectors for co-infection. They allow for the delivery of full-size Cas9 but have lower efficiency due to the need for co-infection of the same cell.	[26]
Lipid Nanoparticles (LNPs)	Lipid nanoparticles are used to deliver mRNA encoding Cas9 and gRNA. They have low immunogenicity risks and are primarily used for liver delivery, although their efficiency in targeting the heart and muscles is limited.	[27,28]
Direct Microinjection (e.g., into Embryos)	This involves the direct injection of CRISPR-Cas9 components (Cas9, sgRNA, and HDR template) into embryos. It is used to correct MYBPC3 mutations in hypertrophic cardiomyopathy models.	[29]
Mitochondrial Localization Signals (MLS)	This enables the targeting of gRNA to mitochondria for mtDNA editing. This method requires specific gRNA modifications to cross mitochondrial membranes.	[30]

**Table 2 cells-14-00131-t002:** Overview of CRISPR-Cas9 applications in cardiovascular diseases.

Disease	Mutation/Target	CRISPR-Cas9 Method Details	Model	Observed Outcomes	References
Familial Hypercholesterolemia (FH)	LdlrE208X (human E207X)	AAV8-mediated CRISPR-Cas9 delivery	In vivo (mouse)	Partial restoration of LDL receptor expression, reduced total cholesterol, LDL cholesterol, triglycerides, decreased macrophage infiltration, smaller plaques, and no significant off-target effects	[51]
Dilated Cardiomyopathy (DCM)	RBM20R634Q, RBM20636s	ABEmax-VRQR-SpCas9 base editing	iPSC-derived cardiomyocytes, mouse	Reversal of cardiac dilation; restoration of normal cardiac function; significant lifespan improvement in treated mice	[53]
Hypertrophic Cardiomyopathy (HCM)	MYH6-R403Q mutation	AAV9-delivered sgRNA targeting MYH6	In vivo (mouse)	Pathogenic allele inactivated in >70% of ventricular myocytes; prevention of structural and functional hallmarks of HCM; normal heart morphology and function	[56,57]
PRKAG2 Cardiac Syndrome	PRKAG2 mutant allele	CRISPR-Cas9 knockout	In vivo (mouse)	Mitigation of ventricular tachycardia and heart failure symptoms	[59,60]
Catecholaminergic Polymorphic Ventricular Tachycardia (CPVT)	RYR2- R176Q mutation	AAV9-CRISPR-Cas9	In vivo (mouse)	Prevention of severe arrhythmia and CPVT onset	[61]
Duchenne Muscular Dystrophy (DMD)	DMD gene exon deletions	AAV9-sgRNA and base-editing CRISPR	In vivo (mouse)	Restoration of dystrophin expression; improved cardiac and skeletal muscle function; halted ventricular remodeling	[63,64,65,66]
Hypertension	AGT, G6PD, GPER1	AAV8-Cas9-AGT gRNA; G6PD ssODN; multiplexed gRNA	In vivo (rat)	Sustained blood pressure reductions; improved arterial elasticity; reduced atherosclerotic risk	[68,69,70,71,72,73,74]
Atherosclerosis	APOC3, PCSK9	Exon 2 disruption, base-editing CRISPR	In vivo (rabbit, hamster, primates)	Lower plasma triglycerides; increased HDL-C; reduced aortic plaque formation; long-term cholesterol reduction in primates	[75,76,77,78]
Mitochondrial Dysfunction	MT-CYB-m.15059G > A mutation	Mitochondrial genome editing; MLS-modified gRNA	Cybrid cell lines	Restoration of mitophagy; reduced pro-inflammatory responses; decreased chronic inflammation linked to atherosclerosis	[82]

## Data Availability

Not applicable.

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
