# Peer review of "CRISPR-Cas9 in Cardiovascular Medicine: Unlocking New Potential for Treatment"

_cells, 2025, doi:10.3390/cells14020131_

Round 1

Reviewer 1 Report

Comments and Suggestions for Authors

The authors provide a detailed review of how CRISPR-Cas9 technology can be applied in cardiovascular medicine. While the manuscript is generally well-written and engaging, several areas require clarification or improvement:

  1. Lines 122–125: The meaning of this paragraph is unclear. Could the authors clarify or rephrase it? Additionally, does this belong under a "Results" section, or is it misplaced?

  2. Sections 3.2–3.3: Several abbreviations are introduced without being fully spelled on their first appearance. Please ensure all abbreviations are defined when first mentioned.

  3. Line 236: What does PLN (hPLN-R14del) refer to? Provide a brief explanation.

  4. Line 255: Clarify the term "A-band-TTNtv."

  5. Line 256: The abbreviation "hiPSC-CMS" should be corrected to "hiPSC-CMs."

  6. Line 280: Explain the term "LZTR1."

  7. Lines 280–282: The conclusion presented here feels abrupt. Could the authors elaborate on how the approach was identified to reverse the pathogenic phenotype in iPSC-CMs derived from Noonan syndrome patients?

  8. Line 436: Revise the sentence “which is comparison is” for grammatical correctness.

Author Response

RESPONSE TO REVIEWER 1 COMMENTS

Dear Reviewer,

Thank you for your thorough and insightful review. We believe your feedback has been invaluable in enhancing the readability and accuracy of our review. In response, we have considered each of your suggestions willingly and attentively. We have compiled a list of all abbreviations used in the manuscript. We would also like to acknowledge that the revised areas are marked in red.

Comment 1 (Lines 122–125): The meaning of this paragraph is unclear. Could the authors clarify or rephrase it? Additionally, does this belong under a "Results" section, or is it misplaced?
Response 1: Thank you for pointing this out. The paragraph in question was mistakenly included and has now been removed from the manuscript.

Comment 2 (Sections 3.2–3.3): Several abbreviations are introduced without being fully spelled on their first appearance. Please ensure all abbreviations are defined when first mentioned.
Response 2: We have expanded and defined all abbreviations upon their first appearance. We hope this improves the clarity and readability of the text.

Comment 3 (Line 236): What does PLN (hPLN-R14del) refer to? Provide a brief explanation.
Response 3: We have included a brief explanation of PLN (hPLN-R14del) in the manuscript to provide clarity.

Comment 4 (Line 255): Clarify the term "A-band-TTNtv."
Response 4: The term "A-band-TTNtv" has been clarified in the manuscript.

Comment 5 (Line 256): The abbreviation "hiPSC-CMS" should be corrected to "hiPSC-CMs."
Response 5: Thank you for catching this error. It has been corrected to "hiPSC-CMs."

Comment 6 (Line 280): Explain the term "LZTR1."
Response 6: We have provided a clear explanation of "LZTR1" in the manuscript.

Comment 7 (Lines 280–282): The conclusion presented here feels abrupt. Could the authors elaborate on how the approach was identified to reverse the pathogenic phenotype in iPSC-CMs derived from Noonan syndrome patients?
Response 7: We have rephrased and elaborated on this section to provide a more comprehensive explanation.

Comment 8 (Line 436): Revise the sentence “which is comparison is” for grammatical correctness.
Response 8: Thank you for noting this. The grammatical error has been corrected.

Reviewer 2 Report

Comments and Suggestions for Authors

The authors present a review on the potential use of CRISPR-Cas9 gene editing technology in clinical applications to treat or prevent CVD. They provide a general overview of the technology and give examples from pre-clinical research that highlight how CRISPR-Cas9 could be beneficial in different types of CVD. They address the important topic of advantages and current challenges of the technology.

While the topic is timely and highly relevant, there are some major concerns with the content and structure of the review that need to be addressed before the article should be considered for publication.

Major comments:

·      Ideas and concepts appear “out of nowhere” and the text just bounces from one topic to the next. This makes the article extremely hard to read. It also is unclear what messages the authors were actually trying to convey. See Line 357 for an Example: The topic “Viruses” suddenly pops up within a paragraph without any connection to the previous section, which was about mutations in the TTN gene. Please structure better (logical sequence of presented topics) and make sure there are transitions between different topics.

·      A table with columns for disease; mutation/target; CRISPR method details; model (eg in vivo mouse or human iPSC culture,…); observed outcomes and literature references should be added to give a better overview to the reader.

·      More focus should be on the bigger picture rather than listing individual studies without much context. For example, a main potential application of CRISPR technology is the repair of gene mutations. Here a section could be added regarding the estimated % of occurrences of each disease with genetic causes that could be addressed by CRISPR vs genetic causes that cannot be addressed or other causes. This would give an idea of the overall potential impact. A different class of applications is where CRISPR can be used similar to an inhibitory drug to dimmish function of certain proteins (eg AGT gene). Concepts like these should be presented more clearly and on a broader level.

·      Specific versions of the Crispr-Cas9 technology are often reference (eg AAV8-Cas9-gRNA) but it is not described well (or at all) what these different versions are. Please check for each occurrence if the specifics are important and either explain or use general terms. Potentially, you could think of creating a table to introduce the different technological details and abbreviations. If not essential for the review article but you still consider it “nice to have” information, write the text using generally understandable terms and only add the method specifics in brackets.

·      Chapter 5: The topic of limitaions and challenges is crucial and has been brushed over too quickly. Several other limitations - besides the ones mentioned - exist (and some are even outlined in the abstract and summary): e.g. genomic stability; precise delivery to a target organ; tissue penetration/ limited efficiency in certain tissues particularly the heart; immune responses; long-term effects and monitoring; complex regulatory frameworks varying between countries. Please extend this section to be more complete.

Minor Comments

·      There are a few editing mistakes (eg 3. Results in row 122-125)

·      I am unsure why gRNA and sgRNA are both used in the text and if they mean the identical or different items. Please clarify.

·      Most of section 110 to 121 is redundant with the following section. Please combine. Between lines 110 and lines 157 the basic principle of  CRISPR-Cas9 is described about 4 times. Any explanation regarding why there are so many repeats is missing. Is there supposed to be a special focus or difference in details between the repeats? If yes, please make this clearer. Currently, it reads as if multiple authors contributed text parts with checking how they fit together.

·      Line 165: There needs to be a logical link between the 2 sentences.

·      Lines 169-170: “produced in 2024 using BioRender. made in 2024 using BioRender.” Redundant. Please delete one.

·      Figures: All figures need to be referenced in the text.

·      Figure captions: 1) This is more of an editing issue but it is extremely unclear what is a figure caption and what is part of the regular text. 2) Please avoid descriptions such as “this figure shows…”. In a figure caption, it is automatically assumed that the text describes what the figure shows.

·      173-192: all of this should be part of the general introduction. (181-185: already in the introduction. Can be deleted)

·      Line 210: The transition between APOC3 and PSCK9 is missing. At lease make a line brea to indicate a new topic.

·      Line 216: Here a transition to the new aspect “PuPGEA” is missing.

·      Line 225: Since the issue of risk associated with germline editing is mentioned here, they concept should have been introduced in an earlier section.

·      Line 233: Abbreviation AAV (and AAV9) not introduced.

·      Line 239: This is not a complete sentence.

·      Line 245: DCM abbreviation needs to be introduced at first occurrence in text.

·      Line 246: “the RBM20R634Q and RBM20636s” This sound like everyone should know what these specific mutations are. Either introduce the specific mutations or describe in more general terms. (the same applies to later parts of the text eg TTNtv)

·      Line 250: TTN: add (Titin).

·      Line 250-254: These 3 sentences are pretty much redundant. Please condense.

·      Line 279: “in patients” is not correct. The study was preclinical in patient-derived cell lines.

·      Line 288: It would be helpful to first introduce the pathophysiological connection between AGT and hypertension before presenting it as a potential target for treatment.

·      Line 293+: same as previous comment but for G6PD.

·      Line 302: same as previous comment but for GPER1.

·      Line 301: G6PD deficiency is also associated with haematological disease.

·      Line 309: UGI abbreviation not explained.

·      Lines 309-319: I strongly suggest to first explain disease mechanisms and causes (underlying mutations) and THEN follow up with how researchers tried to repair those defects using CRISPR-Cas9 technology.

·      Line 322: “.” missing: intervention Un-

·      Lines 324-326: Add some number to show how relevant this aspect is. eg. what % of CVD cases are associated with genetic defects of the mitochondria?

·      Lines 334-358: The topic of macrophage and inflammation role is discussed in a lot of detail (much more detail than other studies included in this review). It is unclear why this topic gets so much more coverage. Please keep depth of the topics consistent – unless this topic is indeed much more relevant than others. In this case please highlight why.

·      Mitochondrial genome editing: references 9 and 71 are both from the same group and on similar topics. Are these the only relevant existing studies? If yes, it needs to be highlighted that this field is in its infancy and virtually no data exists. If there are other groups also working on mitochondrial genome editing in the context of cardiovascular disease, please include more examples.

·      Chapter 5: While the title states “advantages and challenges for clinical applications”, advantages are only presented from a lab-research perspective. Please provide additional information (and references) on expected advantages particular to the clinical setting.

·      Figure 2: “(dis)advEntage”à (dis)advantage

·      Chapter 6: For most of the presented studies it is unclear why they are included as “future or emerging” research directions. The previous chapters had very similar studies presented as existing examples. Please make a clearer distinction between what you deem “future and emerging” vs “state-of-the-art”. (This chapter does not necessarily need to include specific studies since obviously future studies have not been performed yet. It could focus on the next steps envisioned in the papers cited earlier.

·      Line 445: Based on the before provided AGT gene example, CRISPR-Cas9 editing is not only relevant for genetic mutations.

Author Response

RESPONSE TO REVIEWER 2 COMMENTS

Dear Reviewer,

Thank you for your thorough and insightful review. We believe your feedback has been invaluable in enhancing the readability and accuracy of our review. In response, we have considered each of your suggestions willingly and attentively. We have compiled a list of all abbreviations used in the manuscript. We would also like to acknowledge that the revised areas are marked in red.

Major Comments:

Comment 1: Ideas and concepts appear “out of nowhere” and the text just bounces from one topic to the next. This makes the article extremely hard to read. It also is unclear what messages the authors were actually trying to convey. See Line 357 for an Example: The topic “Viruses” suddenly pops up within a paragraph without any connection to the previous section, which was about mutations in the TTN gene. Please structure better (logical sequence of presented topics) and make sure there are transitions between different topics.
Response 1: Thank you for this suggestion. We have restructured significant portions of the text to improve the logical flow between paragraphs. Additionally, we have merged Chapters 3 and 4 into one cohesive section to streamline the discussion and ensure thematic clarity. Redundant sections have been removed in line with your feedback.

Comment 2: A table with columns for disease; mutation/target; CRISPR method details; model (e.g., in vivo mouse or human iPSC culture, …); observed outcomes and literature references should be added to give a better overview to the reader.
Response 2: A table with the requested information has been added to the manuscript to provide a clear and concise overview.

Comment 3: More focus should be on the bigger picture rather than listing individual studies without much context. For example, a main potential application of CRISPR technology is the repair of gene mutations. Here a section could be added regarding the estimated % of occurrences of each disease with genetic causes that could be addressed by CRISPR vs genetic causes that cannot be addressed or other causes. This would give an idea of the overall potential impact. A different class of applications is where CRISPR can be used similar to an inhibitory drug to diminish function of certain proteins (e.g., AGT gene). Concepts like these should be presented more clearly and on a broader level.
Response 3: We have added statistical data in Subsection 3.1, which we created in response to your suggestion. This new section, titled “Overview of CRISPR-Cas9 in Cardiovascular Diseases,” provides a broader context for the application of CRISPR technology. Additionally, we have reorganized Chapter 3 to present applications of CRISPR based on categories, making the information more valuable and accessible to readers.

Comment 4: Specific versions of the CRISPR-Cas9 technology are often referenced (e.g., AAV8-Cas9-gRNA), but it is not described well (or at all) what these different versions are. Please check for each occurrence if the specifics are important and either explain or use general terms. Potentially, you could think of creating a table to introduce the different technological details and abbreviations. If not essential for the review article but you still consider it “nice to have” information, write the text using generally understandable terms and only add the method specifics in brackets.
Response 4: Thank you for this observation. We have added explanations in the text and included a table describing delivery systems used in CRISPR-Cas9 applications. This addition aims to improve clarity and provide readers with a valuable reference.

Comment 5: Chapter 5: The topic of limitations and challenges is crucial and has been brushed over too quickly. Several other limitations - besides the ones mentioned - exist (and some are even outlined in the abstract and summary): e.g., genomic stability; precise delivery to a target organ; tissue penetration/limited efficiency in certain tissues, particularly the heart; immune responses; long-term effects and monitoring; complex regulatory frameworks varying between countries. Please extend this section to be more complete.
Response 5: We have significantly expanded Chapter 5 to address the additional limitations and challenges you outlined.

Minor Comments:

Comment 1: There are a few editing mistakes (e.g., Lines 122–125).
Response 1: The paragraph was mistakenly included and has now been removed.

Comment 2: I am unsure why gRNA and sgRNA are both used in the text and if they mean the identical or different items. Please clarify.
Response 2: We have clarified this terminology in Chapter 2.

Comment 3: Most of Section 110 to 121 is redundant with the following section. Please combine.
Response 3: We have removed redundant parts and consolidated the sections for clarity.

Comment 4: Line 165: There needs to be a logical link between the two sentences.
Response 4: A significant portion of the text has been rephrased to ensure logical continuity.

Comment 5: Lines 169–170: “produced in 2024 using BioRender. made in 2024 using BioRender.” Redundant. Please delete one.
Response 5: Thank you, this redundancy has been corrected.

Comment 6: Figures: All figures need to be referenced in the text.
Response 6: All figures are now referenced in the text.

Comment 7: Figure captions: Clarify what is part of the caption versus regular text. Avoid descriptions such as “this figure shows….”
Response 7: We have revised all figure captions for clarity and conciseness.

Comment 8: Lines 173–192: These should be part of the general introduction.
Response 8: The relevant portions have been moved to the introduction, and redundancies have been removed.

Comment 9: Line 210: Transition between APOC3 and PCSK9 is missing.
Response 9: The text has been revised to ensure logical continuity.

Comment 10: Line 216: Transition to the new aspect “PuPGEA” is missing.
Response 10: The text has been revised to provide appropriate transitions.

Comment 11: Line 225: Since the issue of risk associated with germline editing is mentioned here, the concept should have been introduced earlier.
Response 11: This topic is now introduced in the introduction.

Comment 12: Line 233: Abbreviation AAV (and AAV9) not introduced.
Response 12: We have added explanations, including this term, in Table 1.

Comment 13: Line 239: This is not a complete sentence.
Response 13: The sentence has been rephrased.

Comment 14: Line 245: DCM abbreviation needs to be introduced.
Response 14: The abbreviation has been defined.

Comment 15: Line 246: Introduce or generalize specific mutations.
Response 15: Explanations have been added.

Comment 16: Line 250: Add “Titin” after TTN.
Response 16: This has been done.

Comment 17: Line 250–254: These sentences are redundant.
Response 17: Redundant parts have been removed and rephrased.

Comment 18: Line 279: “In patients” is incorrect.
Response 18: This has been corrected to specify “patient-derived cell lines.”

Comment 19: Lines 324–326: Add statistics on % of CVD cases related to mitochondrial defects.
Response 19: Statistical data has been added in Subsection 3.1.

Comment 20: Chapter 6: Clarify distinction between “future and emerging” and “state-of-the-art.”
Response 20: We have revised Chapter 6 to emphasize exploratory aspects of emerging research while distinguishing it from validated examples in earlier chapters.

Comment 21: Line 445: CRISPR-Cas9 relevance for non-genetic mutations needs clarity.
Response 21: This section has been rephrased for clarity.

Round 2

Reviewer 1 Report

Comments and Suggestions for Authors

The authors very much improved the manuscript.

Author Response

We sincerely thank the Reviewer for their kind words and previous insightful comments, which have greatly contributed to improving the quality of our manuscript. We deeply appreciate the time and effort invested in providing valuable feedback throughout the review process.

Reviewer 2 Report

Comments and Suggestions for Authors

Overall, the manuscript has been substantially improved making it much clearer and easier to read. The train of thought is now a lot more logical.

Minor Comments:

General: The same term for the general technology should be used throughout the article: You switch between CRISPR/Cas9, CRISPR-Cas9, CRISPR and other variations.

Line 60-61: Statement needs a reference. (Typically, CVD-associated mortality is reported to be on the rise.)

Figure 1: This figure makes it seem like the envisioned way of CRISPR/Cas9 therapies is always to modify cells ex vivo and re-apply to the patient. In reality, this is only one option, while direct in vivo treatment of the tissue is another (more relevant!) option. This should be reflected in the figure and/or text and caption.

Lines 243-267: 1) It is doesn’t seem like any of the described CRISPR/Cas9 treatment approaches (Lines 259+) are associated with the mentioned mutations in LDLR gene. Are there CRISPR/Cas9 studies on fixing the LDLR gene? If yes, please specify. If not, this aspect is not relevant to the results. (It can still be mentioned in the introduction or outlook as a potential application for CRISPR/Cas9.)

2) It is unclear if a PCSK9 mutation was corrected/disrupted using CRISPR/Cas9. If yes, please clarify. If not, the approach described in Lines 259+ fits better with the section starting in line 351 (Reducing Protein Expression). In that case also check for redundence with section 388+ and combine as you see fit.

Lines 333+: This seems to belong more to section 298-308 since hypertrophic cardiomyopathy was treated in iPSC – not in embryos or zygotes.

Line 324-332: Since the emphasis here is not so much the disease but rather the early treatment, I suggest putting it at the end of this section after DMD. Alternatively, it can be grouped with the other hypertrophic cardiomyopathy examples (starting line 298).

Line 377: “sex-independent” has not been mentioned as a relevant factor before. This appears out of nowhere and it is unclear how the rat study is “further supporting” this approach since it was not introduced earlier. Maybe rephrase?

Author Response

Response to Reviewer 2 Comments

Dear Reviewer,

Thank you very much for your valuable comments and constructive feedback. Your insights have greatly contributed to improving the clarity, coherence, and overall quality of our manuscript. Please note that all newly introduced changes are marked in red.

  1. The same term for the general technology should be used throughout the article: You switch between CRISPR/Cas9, CRISPR-Cas9, CRISPR, and other variations.

A: We have standardized the terminology throughout the manuscript by consistently using "CRISPR-Cas9" to refer to the general technology.

  1. Line 60-61: Statement needs a reference. (Typically, CVD-associated mortality is reported to be on the rise.)

A: We have rephrased the statement to make it more accurate. Additionally, we have included an appropriate reference to support the revised statement. The new reference addresses the recent trends in CVD-associated mortality and confirms the increase in reported cases globally.

  1. Figure 1: This figure makes it seem like the envisioned way of CRISPR/Cas9 therapies is always to modify cells ex vivo and re-apply to the patient. In reality, this is only one option, while direct in vivo treatment of the tissue is another (more relevant!) option. This should be reflected in the figure and/or text and caption.

A: We appreciate this observation and have updated Figure 1 to illustrate both in vivo and ex vivo approaches to CRISPR-Cas9 therapies. The corresponding figure caption has also been revised to ensure that both methods are clearly described.

  1. Lines 243-267: It doesn’t seem like any of the described CRISPR/Cas9 treatment approaches (Lines 259+) are associated with the mentioned mutations in the LDLR gene. Are there CRISPR/Cas9 studies on fixing the LDLR gene? If yes, please specify. If not, this aspect is not relevant to the results. (It can still be mentioned in the introduction or outlook as a potential application for CRISPR/Cas9.)

A: Thank you for this comment. We have reviewed the literature and included relevant study that specifically address the correction of LDLR mutation using CRISPR-Cas9.

  1. It is unclear if a PCSK9 mutation was corrected/disrupted using CRISPR/Cas9. If yes, please clarify. If not, the approach described in Lines 259+ fits better with the section starting in line 351 (Reducing Protein Expression). In that case also check for redundance with section 388+ and combine as you see fit.

A:We have clarified that CRISPR-Cas9 has been used to disrupt PCSK9 protein levels, with the aim of reducing cholesterol levels. To improve the manuscript’s coherence, we moved the relevant section to the chapter on "Reducing Protein Expression" and combined it with content from Section 388+ to avoid redundancy.

  1. Lines 333+: Point 5: This seems to belong more to section 298-308 since hypertrophic cardiomyopathy was treated in iPSC – not in embryos or zygotes.

A: We agree with this suggestion and have moved this section to 298-308 as proposed.

  1. Lines 324-332: Point 6: Since the emphasis here is not so much the disease but rather the early treatment, I suggest putting it at the end of this section after DMD. Alternatively, it can be grouped with the other hypertrophic cardiomyopathy examples (starting line 298).

A: We have grouped this section with other examples of hypertrophic cardiomyopathy, as you suggested.

  1. Line 377: Point 7: “Sex-independent” has not been mentioned as a relevant factor before. This appears out of nowhere and it is unclear how the rat study is “further supporting” this approach since it was not introduced earlier. Maybe rephrase?

A: We appreciate your attention to this detail. We have rephrased this part to avoid introducing confusion.